# Peer review of "A Physician-Driven Patient Safety Paradigm: The “Pitfall Bank” as a Translational Mechanism for Medical Error Prevention"

_healthcare, 2025, doi:10.3390/healthcare13172248_

Round 1
Reviewer 1 Report
Comments and Suggestions for Authors
Please find the attachment.

Reviewer 2 Report
Comments and Suggestions for Authors
Thank you kindly for the opportunity to review your manuscript concerning the reporting of pitfalls in health care.
I have several comments to share and an overriding question.
Because this manuscript draft does not have line numbers, I will attach a copy of the manuscript mark-up to assure that you can locate in your work my comments. I would ask, for myself and all future reviewers, please consider numbering the lines of your drafts to help communication.
There is an issue that exists throughout this manuscript. It is the use of pronouns that are not clear. Many grammar issues can be corrected by a dedicated editor. In this case, I am concerned that the editor may not know to whom or to what the pronouns refer.
On page 1 the use of the word "it" is problematic as I don't know what it is
On page 2 again the use of "it"
On page 7 "them" and "it"
On page 9 "it" is used twice and confusing twice
There is an inconsistency in the numbering of references. If you look at the references starting on page 13, you will see that the 1st reference is numbered with an Arabic number 1. In the body of the manuscript it is numbered with a Roman numeral 1 on page 2. The rest of the references use Roman numerals until the final reference which uses Arabic number 34, but in the body of the manuscript uses the Roman numeral xxxiv on page 9.
While I understand that this is an opinion / recommendation manuscript, there are some choices of phrasing that seem almost glib. Please review them to assure that they do meet your intent.
"mere carelessness" on page 2. You are discussing an error that could end someone's life. Is "mere" the word you want to choose?
At the bottom of page 2 we see the 1st use of the phrase "goldmine". Is this what you want? Does it clearly describe what is the mine and what is the gold being sought? Goldmines appear again on page 6, but are perhaps more confusing here than when the phrase is previously used.
I am not familiar with infamous graveyards of patients. These references appear on page 7 and are concerning. I believe you may be referring to patients who have died as a result of a medical error. There is no way to use a phrase to describe them that respectfully lessens the burden of killing someone or at least contributing to their death.
On page 8 the phrase IMPASSABLE DIVIDE is used. But on page 9 a description of "passing" that divide is proposed. Is it impassable or not?
There needs to be a careful description of why health care is not science. I believe that most practitioners and researchers know that health care is science. To use these terms as mutually exclusive is problematic and may not be reflecting what you intend to say.
The introduction to point 3 on page 10 has a grammar problem
Consider the following: The inability to identify and solve problems in a timely manner
There is an assumption made on page 10 that needs to be more completely described. The assumption is that the physician can identify the pitfall without creating an error and will have access to the Pitfall Bank. If the physician can identify a pitfall, why does that physician fall into it anyway? Is it all the result of instinct, hubris and experience? If so, what will change that to prevent the error and encourage the use of a Pitfall Bank?
My question focuses on the statement that physicians don't look up solutions because the search is too time consuming. This proposal is that physicians will look up solutions when the solutions are organized more formally and are more easily accessible. I believe that is probably true. If you can provide a tool to make something easier, it will be used. What I do not understand and what has not been fully explained is the identification of the problem or the "pitfall". Are you advocating that a physician look up every step in a medical encounter to see if anyone has ever identified a pitfall? Or are you saying that a physician will create an error even when that physician knows they are at a potential pitfall, but finding an answer is simply too time consuming? Expansion on this would be helpful to the reader

Please see above or look for highlights in the attached markup copy
Round 2
Reviewer 1 Report
Comments and Suggestions for Authors
The authors have responded to my individual comments and improved the manuscript. However, from a reader’s perspective, the paper still remains difficult to follow. Please refer to the points below, and revise the manuscript by summarizing and retaining only the essential content:
- The manuscript does not follow a standardized structure (Introduction, Methods, Discussion, Conclusion).
- The text and tables are presented in an overly list-like manner.
- The Introduction, as well as other sections, are not sufficiently supported by references.
Author Response
Point-by-Point Responses to Reviewer Comments
Response to Comment 1: Insufficient Referencing
Reviewer's Comment: "The Introduction, as well as other sections, are not sufficiently supported by references."
Authors' Response:
We sincerely thank the reviewer for this crucial observation. We fully agree that robust and comprehensive citation is fundamental to building reader trust, substantiating the paper's claims, and accurately situating our work within the broader scientific and policy landscape. This is especially critical in the introduction, which frames the central problem the manuscript seeks to address.
In accordance with this feedback, we have conducted a thorough review of the manuscript and have significantly expanded the referencing throughout. Below, we detail the key assertions that have been newly supported or further strengthened with additional citations.
- I. Substantiating the Scale of Patient Safety Initiatives: The original claim that "healthcare systems worldwide have invested vast resources into a wide array of patient safety initiatives" is now supported with a citation to the 2024 review of patient safety from the UK's Department of Health and Social Care, which provides direct evidence of significant investment and deployed resources.
- II. Corroborating the Lack of Progress in Medical Error Prevention: The assertion of a persistent lack of progress, previously supported by a review of literature up to 2021, has been updated with the same 2024 UK government review. This source confirms that patient safety has not seen significant improvement, demonstrating that the problem is an ongoing, contemporary challenge.
- III. Documenting the Widespread Implementation of Safety-I Tools: The previously uncited claim that the Safety-I approach led to "widespread implementation of standardized tools such as surgical checklists, computerized physician order entry (CPOE) systems, and rigid administrative protocols" is now substantiated with a foundational report from the U.S. Agency for Healthcare Research and Quality (AHRQ), which details the significant federal investment in and promotion of these tools.
- IV. and V. Citing the Limited Effectiveness of Safety-I Tools in Complex Domains: The critical argument that standardized measures "have proven markedly less effective in the complex, dynamic, and cognitively demanding domains of diagnosis and treatment" is now supported by two new citations. These include a seminal critique by Catchpole & Russ (2015) on "The problem with checklists" and a 2024 systematic review by Paterson et al. on the persistent barriers to checklist implementation. These sources provide a nuanced, evidence-based critique of why such tools underperform in complex clinical environments.
- VI., VII., and VIII. Substantiating Key Concepts of the Safety-II Paradigm: Foundational concepts of the Safety-II framework, including the distinction between "work-as-imagined" and "work-as-done" and the shift from reactive prevention to proactive capacity enhancement, are now properly attributed to seminal and contemporary sources, including foundational work by Erik Hollnagel and recent meta-syntheses on healthcare resilience.
In addition to these specific examples, we have updated and added 17 other citations throughout the manuscript to ensure that all assertions are robustly supported. We are confident that these comprehensive additions have fully addressed the reviewer's concern and have significantly improved the scholarly foundation of our paper.
Response to Comment 2: Presentation of Tables and Figures
Reviewer's Comment: "The text and tables are presented in an overly list-like manner."
Authors' Response:
We thank the reviewer for this feedback on the manuscript's presentation. To ensure full conformity with the journal's standards and to improve the clarity of our data presentation, we have carefully reviewed the "Instructions for Authors" regarding the formatting of tables and figures.
Based on this review, we have made the following specific changes to the manuscript:
- Sequential Numbering: All figures and tables have been renumbered sequentially in their order of appearance throughout the manuscript (e.g., Figure 1, Table 1, Figure 2, Table 2, etc.), as stipulated by the journal's guidelines.
- Standardized Caption Placement: We have standardized the placement of all captions in accordance with established academic convention. Captions for tables are now placed directly above the table, while captions for figures are placed directly below the figure.
We believe these adjustments address the reviewer's concern and ensure that the manuscript's formatting is professional, clear, and fully compliant with the journal's requirements.
Response to Comment 3: Non-Standard Manuscript Structure
Reviewer's Comment: "The manuscript does not follow a standardized structure (Introduction, Methods, Discussion, Conclusion)."
Authors' Response:
We appreciate the reviewer's observation regarding the manuscript's structure. We would like to respectfully clarify the rationale for our organizational approach, which we believe is the most effective for the specific type and purpose of this article.
Our manuscript was intentionally submitted under the journal's "Perspective" article category. The primary contribution of this work is not the presentation of new empirical data, which would naturally follow the conventional IMRAD (Introduction, Methods, Results, and Discussion) structure. Instead, its central purpose is to propose and analyze a novel conceptual model for medical error prevention—arguing for a paradigm shift from Safety-I to Safety-II and presenting the "Pitfall Bank" as a translational mechanism to operationalize this shift.1 The current structure, which builds a logical argument from problem analysis to proposed solution, was deliberately chosen to best serve this forward-looking, conceptual goal.
To ensure that this structure is appropriate for the journal, we conducted a review of other "Perspective" articles published in "Healthcare." This review confirmed that the journal allows for considerable structural flexibility for this article type. For example, recent "Perspective" articles by Yun & Zhang (2025), Lisiecka et al. (2025), and Agarwal & Friedman (2025) all deviate from the IMRAD format in favor of thematic or part-based structures that best support their specific argumentative goals.1
Our manuscript, with its progression from a critique of the prevailing paradigm (Safety-I) to an exploration of an alternative (Safety-II), an empirical justification (the bibliometric analysis), and the proposal of a solution (the "Pitfall Bank"), follows this established practice for "Perspective" articles. Given this context, we hope the reviewer will agree that the current structure is not only consistent with the journal's standards but is also the most logical framework for conveying the paper's central conceptual contribution.
Response to the Overarching Comment on Readability and the Need for Summarization
Reviewer's Comment: "The authors have responded to my individual comments and improved the manuscript. However, from a reader’s perspective, the paper still remains difficult to follow. Please refer to the points below, and revise the manuscript by summarizing and retaining only the essential content."
Authors' Response:
We sincerely thank the reviewer for this thoughtful feedback and share the concern for the manuscript's clarity and accessibility to the reader. The point about readability is well-taken, and we have reflected carefully on the balance between detail and conciseness.
The central aim of this manuscript is to introduce a new perspective on the problem of medical errors. This novelty inevitably entails the use of concepts and ideas—such as the Safety-II paradigm and the "Pitfall Bank" as a translational engine—that may be unfamiliar to a broad clinical audience. Consequently, we felt a responsibility to provide more extensive commentary, justification, and illustrative examples than might be necessary for a more conventional topic. Our intention was to ensure that these potentially challenging ideas were not merely stated but were thoroughly explained to prevent misunderstanding and to help the reader fully grasp their implications.
This approach was also informed by our decision to submit to "Healthcare," an open-access MDPI journal. We were motivated by the journal's commitment to reaching a wide and diverse readership, which includes not only specialists in patient safety but also frontline clinicians, healthcare administrators, trainees, and policymakers who may be less familiar with the specific theoretical debates in safety science. The detailed exposition in the manuscript is a deliberate effort to bridge this knowledge gap and make the proposed paradigm shift comprehensible and compelling to this broader audience.
We do, however, agree with the reviewer that a more concise, "summarized" presentation of the core ideas is valuable. We believe such a format is perfectly suited for future abstracts and presentations at scientific conferences, and we intend to disseminate the key concepts in that manner. The purpose of this foundational journal article, however, is to serve as the complete, definitive, and fully reasoned exposition of the proposed model, providing the necessary depth for colleagues to critically evaluate, debate, and build upon our work.
We respectfully ask the esteemed reviewer to consider the manuscript's detail from this pedagogical and mission-oriented perspective. We have taken this opportunity to refine the language throughout the manuscript, aiming for a balance of scholarly precision and clarity that we hope will make the paper's novel concepts more accessible without sacrificing the necessary explanatory depth.1
Summary of Additional Revisions and Compliance Statement
In addition to the major revisions detailed above, we have taken this opportunity to further enhance the manuscript. We have conducted a full proofread of the text to correct minor typographical errors and improve phrasing for clarity and flow.
Furthermore, we have meticulously verified that all figures, schemes, and tables fully comply with the journal's "Instructions for Authors." Specifically, we confirm the following :
- Schemes have been prepared in JPEG format, and one is in color.
- Figures and schemes contain no editable parts.
- All table columns have an explanatory heading.
- All text within figures is in English.
- Figures do not contain unnecessary marks such as red wavy lines or hard/soft returns.
- No special characters or icons are used in any image.
All figures, schemes, and tables included in the manuscript were created by the authors, who accordingly hold the copyright for these materials.1 These additional steps were taken to ensure the final manuscript is as polished and submission-ready as possible.
Concluding Remarks
We once again thank the editor and the reviewers for their insightful and valuable guidance. The peer-review process has been instrumental in helping us refine our arguments, strengthen our evidence base, and improve the overall quality of our manuscript. We hope that the revisions we have made fully address all concerns and that the manuscript is now deemed suitable for publication.
Sincerely,
The Authors
Reviewer 2 Report
Comments and Suggestions for Authors
I am truly impressed by this version of your manuscript. I appreciate the time and effort you contributed to making this presentation of your information appropriate for all health care professionals.
Author Response
We once again thank the editor and the reviewers for their insightful and valuable guidance. The peer-review process has been instrumental in helping us refine our arguments, strengthen our evidence base, and improve the overall quality of our manuscript. We hope that the revisions we have made fully address all concerns and that the manuscript is now deemed suitable for publication.
Sincerely,
The Authors